# Using Out-of-Batch Reference Populations to Improve Untargeted Metabolomics for Screening Inborn Errors of Metabolism

**DOI:** 10.3390/metabo11010008

**Published:** 2020-12-25

**Authors:** Michiel Bongaerts, Ramon Bonte, Serwet Demirdas, Edwin H. Jacobs, Esmee Oussoren, Ans T. van der Ploeg, Margreet A. E. M. Wagenmakers, Robert M. W. Hofstra, Henk J. Blom, Marcel J. T. Reinders, George J. G. Ruijter

**Affiliations:** 1Department of Clinical Genetics, Erasmus Medical Centre, Dr. Molewaterplein 40, 3015 GD Rotterdam, The Netherlands; r.bonte@erasmusmc.nl (R.B.); s.demirdas@erasmusmc.nl (S.D.); e.jacobs@erasmusmc.nl (E.H.J.); r.hofstra@erasmusmc.nl (R.M.W.H.); h.j.blom@erasmusmc.nl (H.J.B.); 2Department of Pediatrics, Center for Lysosomal and Metabolic Diseases, Erasmus Medical Centre, Dr. Molewaterplein 40, 3015 GD Rotterdam, The Netherlands; e.oussoren@erasmusmc.nl (E.O.); a.vanderploeg@erasmusmc.nl (A.T.v.d.P.); 3Department of Internal Medicine, Center for Lysosomal and Metabolic Diseases, Erasmus Medical Centre, Dr. Molewaterplein 40, 3015 GD Rotterdam, The Netherlands; m.wagenmakers@erasmusmc.nl; 4Faculty of Electrical Engineering, Mathematics and Computer Science, TU Delft, Van Mourik Broekmanweg 6, 2628 XE Delft, The Netherlands; M.J.T.Reinders@tudelft.nl

**Keywords:** untargeted metabolomics, inborn errors of metabolism, normalization, internal standards, batch effects

## Abstract

Untargeted metabolomics is an emerging technology in the laboratory diagnosis of inborn errors of metabolism (IEM). Analysis of a large number of reference samples is crucial for correcting variations in metabolite concentrations that result from factors, such as diet, age, and gender in order to judge whether metabolite levels are abnormal. However, a large number of reference samples requires the use of out-of-batch samples, which is hampered by the semi-quantitative nature of untargeted metabolomics data, i.e., technical variations between batches. Methods to merge and accurately normalize data from multiple batches are urgently needed. Based on six metrics, we compared the existing normalization methods on their ability to reduce the batch effects from nine independently processed batches. Many of those showed marginal performances, which motivated us to develop *Metchalizer*, a normalization method that uses 10 stable isotope-labeled internal standards and a mixed effect model. In addition, we propose a regression model with age and sex as covariates fitted on reference samples that were obtained from all nine batches. *Metchalizer* applied on log-transformed data showed the most promising performance on batch effect removal, as well as in the detection of 195 known biomarkers across 49 IEM patient samples and performed at least similar to an approach utilizing 15 within-batch reference samples. Furthermore, our regression model indicates that 6.5–37% of the considered features showed significant age-dependent variations. Our comprehensive comparison of normalization methods showed that our *Log-Metchalizer* approach enables the use out-of-batch reference samples to establish clinically-relevant reference values for metabolite concentrations. These findings open the possibilities to use large scale out-of-batch reference samples in a clinical setting, increasing the throughput and detection accuracy.

## 1. Introduction

The screening of patients suspected for inborn errors of metabolism (IEM) is currently based on measuring panels of specific groups of metabolites, like amino acids or organic acids using a number of different tests, and techniques, such as ion-exchange chromatography, liquid chromatography mass spectrometry (LC-MS) and gas chromatography mass spectrometry (GC-MS). This targeted approach with several different tests is time consuming and limited in the number of metabolites being analyzed. Untargeted metabolomics using high resolution accurate mass liquid chromatography mass spectrometry (HRAM LC-MS) can detect hundreds to thousands of metabolites within one test and, as a consequence, receives increasing interest to be used in IEM screening [1,2,3,4,5]. Moreover, untargeted metabolomics can also reveal new biomarkers or increase our understanding of disease mechanism when exploited in epidemiological studies [6].

In traditional targeted diagnostic laboratory tests, hundreds of reference samples are required for establishing robust reference intervals. When using untargeted metabolomics, the establishment of reference values is complicated, due to the semi-quantitative nature of the data, owing to several sources of variation, like injection volume, retention time, temperature, or ionization efficiency in the mass spectrometer that cannot easily be amended. Moreover, these variations are even larger between different measurement runs in which a batch of samples is being measured simultaneously, hampering the resemblance between different batches. Consequently, the within-batch variation is smaller than between-batch variation. Targeted metabolomics generally deals with these technical variations by using internal standards that are chosen such that they are chemically identical (an isotope) or similar to the metabolite of interest, thereby making it possible to accurately measure its quantity. However, since untargeted metabolomics involves the measurement of a large number of different metabolites/features, it becomes unfeasible to add an internal standard for each metabolite. Therefore, in order to conquer these batch effects, the current approaches include reference samples in each single batch of measurements [1,2,3,4,5] to improve detection sensitivity (due to tighter reference values as a result of lower variation in the within-batch reference samples).

Clearly, this reduces the throughput efficiency of IEM screening, as the number of patient samples that can be included in a batch is considerably lower when reference samples also need to be measured. However, more importantly, the number of reference samples in one batch might fall short in the establishment of adequate reference ranges as variations in certain metabolites are not captured well enough in the relatively small reference panel. For example, factors, like age, sex, and BMI, can affect abundancies of metabolites and, to establish reliable reference ranges, one thus needs to correct for these factors by using a large number of reference samples [7,8,9,10]. Consequently, for reliable untargeted metabolomics in clinical testing, a large set of reference samples is needed, while, for throughput efficiency, a small set is preferred. Altogether, this calls for an approach that can establish reference values that are based on reference samples being measured in several batches (out-of-batch controls).

When relying on reference samples from different batches, one needs to correct for the batch effects in order to obtain reliable estimates for the reference ranges. This is generally solved by normalization methods and some have already been proposed within the context of untargeted metabolomics and mass spectrometry [11,12,13]. Only a few groups have used out-of-batch reference samples to determine the reference values and used relatively simple normalization techniques, like median scaling [1], a reference internal standard per metabolite [3], or anchor samples [6]. However, there has not been an extensive exploration of normalization techniques within the context of diagnostic testing for IEM’s.

We explore several known normalization methods for their ability to remove batch effects and detect biomarkers from patients with known IEM. Furthermore, we introduce a new normalization method, which we called *Metchalizer*, which uses internal standards and a mixed effect model to remove batch effects. As this allows for a large set of (out-of-batch) reference samples, we also explore a regression model that uses age and sex as covariates to correct for potential age and sex effects on the reference values. Using the regression model combined with the *Metchalizer* normalization, we achieve similar performances in biomarker detection as compared to the use of within-batch controls. Hence, this opens the possibility to increase the throughput of untargeted metabolomics in IEM screening as well as including more complex confounder strategies. *Metchalizer* and the regression model are available at https://github.com/mbongaerts/Metchalizer.

## 2. Results

### 2.1. Data and Batch Characteristics

Using ultra-high performance liquid chromatography-Orbitrap-MS (UHPLC-Orbitrap-MS), nine untargeted metabolomics runs/batches were measured containing 261 control samples and 58 IEM patient samples, together having 35 unique IEMs. All nine batches were measured on a single mass spectrometer (Thermo Scientific Q Exactive Plus), while three separate Kinetex F5 columns for ultra-high performance liquid chromatography (UHPLC) were used. Using in-house developed software, features across the nine batches were matched and accordingly merged into a single dataset (see Section 4.2). After merging, 446 positively ionized features were obtained, among which 114 were annotated, and 328 negatively ionized features were attained with 82 annotated features. We only included features that were merged across all nine batches to ensure consistency among the findings. This resulted in the loss of IEM biomarkers, and the full list of the lost biomarkers per IEM can be found in Appendix E. Intra-batch coefficients of variation (CV) on 17 (internal and external) standards were smaller (median CV = 14%, see Appendix A
Figure A1) than inter-batch CV’s (median CV = 27%, see Appendix A
Table A1), indicating that batch effects were present. Principle Component Analysis (PCA) further demonstrated the presence of batch effects, as shown in Figure 1A,B, showing the first three PCs for the raw abundancies (*Raw* and *Log-Raw*).

### 2.2. Comparing Normalization Methods

We investigated the performance of several normalization methods on batch effect removal by evaluating multiple metrics that are based on quantitative measurements, the Quality Control (QC) samples and PCA analysis (Section 4.4.5).

**Reduced batch effects:** from the PCA plots, we observe that most normalization methods reduced batch effects, since batch clustering seemed to be reduced after normalization (Figure 1), which is confirmed when looking at the *batch prediction score* (Figure 2A), showing lower scores for normalized abundancies when compared with the raw data (*Raw* or *Log-Raw*). *BC-Metchalizer*, *Log-Metchalizer* had the lowest *batch prediction scores*, with median scores of 0.12 (0.11), 0.12 (0.12) for positive (negative) ion mode, respectively (see Appendix B
Table A2 for all medians).

**Most methods conserve separation of QC samples:** QC samples were included in every batch and thet were expected to segregate from the human plasma samples in the first four principle components (PC) due to overall abundancy differences for several metabolites (see Figure 1 for the first 3 PCs). Normalization should maintain this separation, which was quantified by the *QC prediction score* (Figure 2B). We observe that, for most normalization methods, the median *QC prediction score* was about 1.00. Although, *Log-NOMIS* scored relatively well when considering the *batch prediction scores*, with a median score of 0.20 (0.17) for positive (negative) ion mode, it performed poor on the *QC prediction score*, with a median of 0.33 (0.88) for positive (negative) ion mode. Therefore, it is likely that this method removed variations other than batch related variation.

**Resemblance with quantitative measurements:** to further quantify batch effect removal, we calculated the *Spearman score* and *$R^{2}$ score* between quantitative plasma concentrations (in μmol/L) and the normalized abundancies of our evaluation set of amino acids and (acyl)carnitines (Section 4.4.5). In order to ensure high signal-to-noise ratio’s in the quantitative measurements, we selected only metabolites having a population average concentration above 1 μmol/L. Matching the evaluation set with the annotated features in the untargeted metabolomics data resulted in 15 and 10 metabolites in positive and negative ion mode, respectively. Figure 2C,D shows both metrics for the investigated normalization methods. Again, for most normalization methods, both of the metrics improved when compared to the raw data (*None-Raw*). *BC-Metchalizer*, *Log-Metchalizer* and *None-Anchor* appeared to perform the best on these metrics with median $R^{2}$ scores of 0.66 (0.64), 0.61 (0.68), 0.63 (0.57), and median *Spearman scores* of 0.78 (0.79), 0.78 (0.83), 0.75 (0.74), for positive (negative) ion mode.

**Reduced between-batch variation in QC samples:** next, we compared the within-batch variance of the QC samples with respect to the total variance which is expressed by the *WTR score* (Section 4.4.5) for each normalization method. *WTR scores* close to 1 indicate the absence of batch effects. *None-Raw* and *Log-Raw* had low WTR scores and after normalizing these scores increased (Figure 2E). *BC-Metchalizer* and *Log-Metchalizer* scored among the highest on this WTR score. *None-Anchor* had high WTR scores, which was expected, since *None-Anchor* uses the QC samples for normalization and, consequently, the *WTR scores* are biased towards higher values.

**Preserved resemblance of QC samples:** removal of variation results in higher WTR scores, but also potentially removes variation(s) of interest. Therefore, we investigated whether the resemblance of all QC samples among each other was conserved after normalization using the Spearman correlation. Lower Spearman correlations indicate that variation of interest might also be lost, since the resemblance between the QC samples is reduced. Figure 2F shows the *QC correlations* for each normalization method. These results show that *Log-Fast Cyclic Loess* and *Log-RUVrand* also removed the non-batch related variations, even while having relatively good *QC prediction scores*.

Additionally, we investigated whether normalization improved the resemblance with patients sharing the same IEM and, likewise, reduced the resemblance between patients having a different IEM. This analysis shows that *BC-Metchalizer* and *Log-Metchalizer* were among the best when considering two different resemblance scores (see Appendix K for more details).

Taken together, *BC-Metchalizer*, *Log-Metchalizer*, and *None-Anchor* showed the optimal normalization characteristics across the evaluation metrics and were evaluated in more detail (see Section 2.4).

### 2.3. Confounder Effects of Age and Sex

We developed a regression model with sex as covariate and age as a polynomial (*p* = 1, 2, 3) covariate in order to explore confounding effects of age and sex on metabolite abundancies (see Section 4.5.2). After normalization, we fitted the model parameters for every feature while using all of the samples that are present in the nine batches and determined the significance of the coefficients in the regression model (Section 4.5.2). The obtained *p*-values were corrected for multiple testing per coefficient and ion mode using the Benjamin–Hochberg procedure (FWER = 0.05). Table 1 shows the percentages of significant coefficients (α=0.05) per ion mode and (selected) normalization method. Our findings suggest that 6.5–37% of all features showed age dependency when looking at coefficient ${\hat{\beta }}_{1}^{\mathrm{Age}}$ (i.e., the linear term in the model). It is noteworthy that more age-related features were found in the negative ion mode.

Although the significance of the regression coefficient indicates whether the determined coefficient is a true finding, the (relative) magnitude of the coefficient determines the effect size. While selecting only significant coefficients ${\hat{\beta }}_{1}^{\mathrm{Age}}$ with an effect size larger than 2% per year (see Appendix C
Figure A3 for explanation), we found that around 1–7% of all features in positive ion mode, and 5–22% of all features in negative ion mode, showed (strong) age-dependency (Appendix C
Table A5). Moreover, age-dependent features have the tendency to increase/decrease in abundancy faster at younger and older ages, which implies that a matching reference population for these age groups are more important (see Appendix C
Figure A2).

When using normalization by *BC-Metchalizer*, age-dependent metabolites (Appendix C
Table A3), include known IEM biomarkers, such as: guanidinoacetic acid(+), homoarginine (−), 2-ketoglutaric acid (−), C3 propionylcarnitine (+), phenylacetic acid (+), and uridine (−). As an example, we plotted the regression model for guanidinoacetic acid (Figure 3), illustrating that the Z-score for a fixed abundancy depends on age (and slightly on sex at later ages). This also shows a non-linear trend with age. Our analyses showed that more metabolites have significant non-linear trends over age (${\hat{\beta }}_{2}^{\mathrm{Age}}$ and ${\hat{\beta }}_{3}^{\mathrm{Age}}$ in Table 1).

No significant gender-related features were found (Table 1) and just 0.5%, 0.7% of all features in positive ion mode showed significant sex-age interaction (${\hat{\beta }}_{}^{\mathrm{Sex},\mathrm{Age}}$), for *None-Anchor* and *BC-Metchalizer*, respectively. Among these features are biomarkers: guanidinoacetic acid(+) and ornithine(−). See Appendix C
Table A4 for more details.

### 2.4. Detection of the Expected IEM Biomarkers

Next, we investigated the impact of normalization and using out-of-batch reference samples on expected biomarker detection in the 49 (/58) IEM patients (see Section 4.6 and Appendix D
Table A6) by plotting the number of detected expected biomarkers (expected true positives) against the average number of positives (true plus false positives) per patient at various Z-score or *p*-value thresholds (Section 4.6), similar to a Receiver Operator Curve (ROC). Untargeted metabolomics did not allow for us to make a distinction between false positives and true positives, due to unannotated features and even unknown disease related features/biomarkers. When assuming that the majority of the positives per patient are false positives, we used the average number of positives per patient as a proxy for the false positives. Improved performance was considered to increase the number of detected expected biomarkers (true positives of which we are certain) while lowering the average number of positives per patient, thereby increasing the Area Under the Curve (AUC) (see Section 4.6 for more explanation).

We decided to take the method that uses 15 within-batch reference samples and raw abundancies (*15in&None-Raw*) as the reference approach. Performance was expressed as a percentage of this reference AUC, as indicated by ${\mathrm{AUC}}_{15\mathrm{in}\&\mathrm{None-Raw}}^{x}$ (where *x* indicates if the AUC was created from the average Z-scores or *p*-values). These *p*-values were obtained from the Welch’s *t*-test, which tests whether the average Z-score of an expected biomarker or feature across the triplicate significantly differs from the average Z-score of the reference population (Section 4.5.4).

**Log-transform improves biomarker detection for *p*-values:** our first observation is that, when considering the Z-scores, the log-transformed raw abundancies (*15in&Log-Raw*) have an AUC approximately equal to ${\mathrm{AUC}}_{15\mathrm{in}\&\mathrm{None-Raw}}^{Z}$ (Figure 4), implying that this transformation hardly affected this performance metric. However, when using the *p*-values, the log-transformation improved the detection of the expected biomarkers, as ${\mathrm{AUC}}_{15\mathrm{in}\&\mathrm{Log-Raw}}^{p}$ is 8% higher than the ${\mathrm{AUC}}_{15\mathrm{in}\&\mathrm{None-Raw}}^{p}$ (Figure 4).

**Reduced performance with age/sex matched out-of-batch references:** when comparing the performance of using 15 out-of-batch samples (*15out&None-Raw*) to the *15in&None-Raw* reference, the performance for *15out* was clearly reduced (Figure 4A), achieving only 84% of the reference ${\mathrm{AUC}}_{15\mathrm{in}\&\mathrm{None-Raw}}^{Z}$. This difference was also present when looking at the *p*-values, which resulted in a clear reduction of the ${\mathrm{AUC}}_{15\mathrm{in}\&\mathrm{None-Raw}}^{p}$ (77%). Hence, the potential improved age/sex matching for *15out*, due to the increased number of available reference samples (Appendix H), did not result in improved performance, most likely due to the dominance of batch effects.

**Normalization improves performance of age/sex matched out-of-batch controls:** after normalizing with *BC-Metchalizer*, *Log-Metchalizer*, or *None-Anchor* and using 15 out-of-batch controls (*15out*), the performance increased when compared to *15out&None-Raw* (Figure 4A–C), and it came closer to ${\mathrm{AUC}}_{15\mathrm{in}\&\mathrm{None-Raw}}^{Z}$; for *BC-Metchalizer* 94%, *Log-Metchalizer* 94%, and *None-Anchor* 93%. Interestingly, when considering biomarker detection performance using the *p*-values, the *BC-Metchalizer* performed on par with *15in&None-Raw* (98%), *Log-Metchalizer* improved over *15in&None-Raw* (104%), while *None-Anchor* was 92%. *Log-Metchalizer* performed similarly to *15in&Log-Raw* (104% and 108%, respectively), indicating that out-of-batch samples can be used instead of within-batch samples to determine reference values.

**Regression model effectively models age and sex effects:** the performance for ${\mathrm{AUC}}_{}^{Z}$ using the regression model (*Regression*) remained the same for all considered normalization methods with respect to *15out*, see also Figure 4A–C. When considering the *p*-values, ${\mathrm{AUC}}_{}^{p}$, the performance was also similar to *15out*; *BC-Metchalizer* (−1%), *Log-Metchalizer* (−1%), *None-Anchor* (−2%) (Figure 4 D–F). Interestingly, when we took all of the reference samples to determine the Z-scores (*All samples*, Methods), similar ${\mathrm{AUC}}_{}^{Z}$ performances were observed when compared to *Regression*, i.e., +0% for *BC-Metchalizer* and +3% *Log-Metchalizer* and +0% for *None-Anchor*. When considering the *p*-values the difference were larger, i.e., −5% for *BC-Metchalizer* and −1% *Log-Metchalizer*, and −6% for *None-Anchor*, suggesting an influence of age- and sex effects on the detection of biomarkers.

## 3. Discussion

Targeted measurements of metabolites in body fluids using various platforms, such as HPLC, GC-MS, and LC-MS/MS are traditionally applied for laboratory diagnosis of IEM. For each individual metabolite, age- and, sometimes, sex-dependent reference ranges are established while using hundreds of reference samples. Untargeted metabolomics is a promising alternative by enabling the determination of many metabolites in one analysis. This can speed up the diagnostic process and extend the number of different IEMs that can be screened in a single run. A major drawback of current approaches is that reference samples need to be included in the same experimental batch in order to ensure proper reference ranges (or Z-score transformations). Some methods do exist that use reference samples that were measured in different batches (out-of-batch samples) to determine age and sex corrected Z-scores, and they are based on methods that remove the technical variations. There has not been a comprehensive comparison of the different normalization methods with approaches that use out-of-batch samples, which we have set out in this work. Moreover, we developed a new normalization method, *Metchalizer*, which makes use of internal standards, an approach that has been shown to be useful when mapping specific metabolites to specific internal standards [3], and that we generalize to all features measured. Because more reference samples are available when using the out-of-batch samples, we additionally propose a regression model that incorporates sex and age effects as (non-linear) covariates. Altogether, we have shown that our methodology has biomarker detection performances that are at least similar to using 15 within-batch samples.

Typically, around 20,000 features in both negative and positive mode were detected per batch. When we require a feature to be measured (and matched) in all nine batches, we retained 446 positively and 328 negatively ionized features, respectively. Because some normalization methods use a statistical approach (*PQN*, *Fast Cyclic Loess*), the reduction in features might explain the reduced performance of these methods. In addition, the requirement of features being measured (and matched) across all nine batches resulted in the loss of clinically relevant biomarkers (see Appendix E), which is a significant limitation of using out-of-batch samples. This suggests that within-batch references are still required when this limitation cannot be conquered. As an alternative, we could have made the inclusion of features dependent on fewer batches (for example, being present/matched in >5 out of 9 batches). We decided not to do that in this study, as this would have resulted in an unequal number of reference samples for the different features, leading to inconsistent results between the out-of-batch methods. The availability of more batches could have solved this issue, because an equal number of reference samples could likely be obtained per feature, even when these features were not present/matched in some batches. It is interesting to note that our proposed normalization method (*Metchalizer*) showed consistent performances when data from a varying number of batches were used (Appendix G). Some biomarkers, for example, isobutyrylglycine, were only detected in the batches containing patient samples with elevated levels of these specific metabolites. We anticipate that, for this kind of biomarkers, out-of-batch strategies are less useful, since abundancies in (normal) references are (very) low, thereby making out-of-batch Z-score calculation unsuitable.

*Anchor* uses anchor (fixed) samples, measured in all batches, in order to normalize the features. *Anchor* normalization on none-transformed data performed well when compared to most of the other normalization methods explored, but slightly less than *BC-Metchalizer* and *Log-Metchalizer* when considering the performance metrics *Spearman score*, *$R^{2}$ score*, *batch prediction score*, and performance on biomarker detection. We anticipate that the anchor samples may not correlate with all types of variation, like, for example, injection volume, which is a source of variation at the sample level, whereas the abundancy of the internal standards (used by *Metchalizer*) is directly linked to the injection volume. *Anchor* also assumes that metabolite levels remain constant over time in the anchor samples. As a consequence, if, for example, storage effects take place, *Anchor* is impeded. The use of *Anchor* may also be less practical, because it requires the same anchor samples in every batch. The introduction of a new anchor sample requires an ‘overlapping batch’ containing a set of both the former anchor sample together with the newly introduced anchor samples.

*Metchalizer* is based on the basic assumption that all variations which can be explained by the variation in the abundancies of the internal standards (being the latent variables from PLS analysis) are technical variations, including batch effects. It exploits the linear relationship between these latent variables and the feature being measured across all samples, thereby capturing the covariance between the standards and that feature. *Metchalizer* assumes that this relationship holds across batches and with that assumption determines (batch) intercepts that correct for ’unexplained’ batch/technical variations. Consequently, *Metchalizer* can correct for large batch effects, but this comes with the potential danger of overcorrection when batches differ from each other due to biological variation, which will then be interpreted as ’unexplained’ batch/technical variations. For this reason, it is important to use randomized samples in each batch (in terms of age, sex, etc.) in order to minimize the possibility of biological variations between batches.

*Log-Metchalizer* log transforms the abundancies before applying *Metchalizer*, whereas the *BC-Metchalizer* uses a less strong Box–Cox transformation. The effect of this stronger transformation on most investigated metrics in this study was small, although we did observe that a stronger initial transformation led to improved biomarker detection performances when considering the *p*-values. *15in&None-Raw* had a lower ${\mathrm{AUC}}^{p}$ than *15in&Log-Raw* and it could therefore also explain the improved performance of *Log-Metchalizer* over *BC-Metchalizer* on this metric. A simulation showed that log-transforming the raw abundancies indeed caused differences in the obtained Z-scores and *p*-values when compared to the raw abundancies (Appendix I). Positive Z-scores had relatively lower *p*-values (and vice versa) for log-transformed abundancies and this could therefore explain the improved performance on biomarker detection, since most of the considered biomarkers had positive Z-scores, thus biasing this performance metric. Increasing the number of internal standards did not improve the normalization performance when considering metrics that are based on the quantitative measurements, although we observed that certain combinations of internal standards improved the normalization of specific metabolites (Appendix F). This suggests that *Metchalizer* might be improved by matching features/metabolites with a certain set of internal standards (for example, based on retention time).

We were a bit surprised that biomarker detection performance while using the Z-scores (${\mathrm{AUC}}_{}^{Z}$) for the regression model was similar to using all of the samples, as abundancies are known to be dependent on age (and sex). Plotting the differences between the obtained Z-scores in a Bland–Altman plot show that, on average, no differences are present between the two approaches; for all features as well as the IEM biomarkers (see Appendix J). This explains why the ${\mathrm{AUC}}_{}^{Z}$ performances are similar for both Z-score approaches, since, for a given Z-score cutoff (used to make the ROC curve), the number of positives is approximately the same, and the same holds when looking at the number of biomarkers detected for a given Z-score cutoff. However, this does not necessarily imply that the regression model is less or equally accurate in determining abberations. We anticipate that the performances for *Regression* would outperform *All samples* when more age-dependent IEM biomarkers were included. Additionally, when judging biomarker detection using the *p*-values, we did see that *Regression* slightly outperformed *All samples*.

## 4. Materials and Methods

### 4.1. Untargeted Metabolomics Datasets

While using UHPLC-Orbitrap-MS, human plasma samples of 261 control samples and 58 IEM patients were measured over nine batches over the period 10 December 2018 to 10 January 2020 [5], having, in total, 35 unique IEMs. In agreement with national legislation and institutional guidelines, all patients or their guardians approved the possible anonymous use of the remainder of their samples for method validation and research purposes. The study was conducted in accordance with the Declaration of Helsinki. For every patient, a technical triplicate was included, which allows for obtaining more certainty regarding the measured abundancy (or Z-score) by looking at the spread in the triplicate. A QC (Quality Control) sample was included in all nine batches and 5–9 technical replicates were present in every batch. Because the QC sample was a commercial sample, the sample differed in the concentration of several metabolites when compared to the (average) concentrations of the human plasma samples that were analyzed in these datasets. Features were annotated, as described in Bonte et al. [5]. For eight batches, 18–40 normal controls have been measured (no triplicate) to ensure more accurate reference values. These controls were random selected samples in terms of age and sex, and none of the samples (patients and controls) were measured in more than one batch. One batch included 22 triplicates measurements (plus QC samples) and no control samples. In this study, we will refer to ‘feature’ as being either a single *m*/*z*-value (with unique retention time) or a merge of multiple features, where the adduct type and/or isotope was determined with corresponding neutral mass and, consequently, merged to a single feature.

The following internal standards have been added to each batch in order to facilitate normalization that is based on these internal standards: 1,3-${}^{15}$N uracil (+/−) [300 µmol/L], 5-bromotryptophan (+/−) [85 µmol/L], D${}_{10}$-isoleucine (+/−) [500 µmol/L], D${}_{3}$-carnitine (+/−) [285 µmol/L], D${}_{4}$-tyrosine (+/−) [230 µmol/L], D${}_{5}$- phenylalanine (+/−) [600 µmol/L], D${}_{6}$-ornithine (+) [225 µmol/L], dimethyl-3,3-glutaric acid (+/−) [300 µmol/L], ${}^{13}$C-thymidine (+/−) [300 µmol/L], D${}_{4}$-glycochenodeoxycholic acid (−) [44 µmol/L], where + indicates positive ion mode, and—indicates the negative ion mode.

### 4.2. Data Processing

Pre-processing steps (alignment, peak picking etc.) were performed per batch while using Progenesis QI v2.4 (Newcastle-upon-Tyne, UK) [5]. In-house software was developed in order to match features from each batch to a reference batch, which, in this case, was the fifth batch when sorting on chronologically order. Chromatograms between batches were initially aligned to the reference batch by using lowess regression, where the features were matched based on retention time difference, *m*/*z*-value, and median abundancy difference similar to the criteria described below.

Matching features was performed based on several criteria:When features were annotated in reference batch and the batch being merged, these features were pooled to the merged dataset.When MS/MS spectra were present for a potential matching pair of features, the cosine similarity metric was calculated and it had to be >0.8.The retention time difference in percentage was calculated between potential matches, and it had to be <3%.Progenesis QI determined per feature an isotope distribution and we required sufficient overlap of these distributions between potential matching pairs. This was determined by calculating a difference in the percentage between each bin of this distribution. The maximum difference of these bins had to be <25%.Despite the batch effects and potential biological differences between batches, we expected the within-batch median of the (raw) abundancies for matching features to be at least similar. We calculated the differences between these medians in percentages, and required that this difference was <300%.When neutral masses were known for the matching pair, but not the MS/MS spectra, the ppm-error had to be <1.When *m*/*z*-values were known for the matching pair, but not the MS/MS spectra and neutral masses, the ppm-error of between the *m*/*z*-values had to be <1.

When a feature from the reference batch had two or more (potential) matches with the batch being merged, we decided to exclude these matches, since it was not clear which match would be the correct one. Similarly, when a feature from the batch being merged had more than one match with the reference batch, this feature would also be excluded. The resulting merged dataset only contained features that were matched (i.e., fulfilling the above matching criteria) across all nine batches. Consequently, this led to the loss of circa 98% of the number of features that were normally detected within the batch.

### 4.3. Quantitative Evaluation Set

For the evaluation of the normalization methods, the following 15 metabolites were quantitatively (µmol/L) measured in two separate assays: leucine (+), C0 L-carnitine (+/−), methionine (+/−), C2 acetylcarnitine (+), 5-aminolevulinic acid/4-hydroxyproline (+), citrulline (+/−), aspartic acid (−), glutamine (+/−), (allo)isoleucine (+/−), proline (+), tyrosine (+), phenylalanine (+/−), taurine (+/−), asparagine (+/−), and arginine (+/−). Amino acids were determined by ion-exchange chromatography according to protocols described by the manufacturer (Biochrom). Free carnitine and acylcarnitines analysis was performed, as described by Vreken et al. [14].

### 4.4. Normalization Methods

#### 4.4.1. Initial Transformations

Prior to normalization, raw abundancies were, for some methods, transformed while using a log-transform or Box–Cox transformation given by $\hat{y}\;=\;\left({\left(y+\lambda_{2}\right)}^{\lambda_{1}}-1\right)/\lambda_{1}$ with $\lambda_{1}\;=\;0.5$ and $\lambda_{2}\;=\;1$. If an initial transformation was applied, this was indicated in the name of the (normalization) method, where ‘BC-’ refers to the Box–Cox transformation and ‘Log-’ to the log transformation. When no transformation was performed, this was indicated with ‘None-’.

#### 4.4.2. Normalization by Metchalizer

Metchalizer assumes a linear mixed effect relationship between the abundancies of the internal standards and the feature of interest. Because the internal standards were expected to be correlated, we represented them by an orthogonal set of covariates. These covariates are obtained as the Latent Variables (LV) from the Partial Least Squares (PLS) of the set of internal standard abundancies (represented in matrix X) and the (categorical) information regarding which sample belonged to which batch (represented by matrix Y). The number of LV’s were chosen from the metric I(K):(1)$$I\left(K\right)\;=\;\sum_{k=1}^{K}\sum_{b,i}{\left(x_{ib}^{{\mathrm{LV}}_{k}}-{\bar{x}}_{ib}^{{\mathrm{LV}}_{k}}\right)}^{2}$$
where ${\bar{x}}_{ib}^{{\mathrm{LV}}_{k}}$ is the center of batch *b* in the direction of ${\mathrm{LV}}_{k}$. We selected that *K*, for which I(K) reached 75% of its maximum value.

The mixed effect model then considers the LV’s as fixed effects and all variations not explained by the LV’s are considered as (random) batch effects:(2)$${\hat{y}}_{ijb}\;=\;\beta_{j}^{0}+\sum_{k=1}^{\mathrm{selected}\;K}\beta_{j}^{k}x_{i}^{{\mathrm{LV}}_{k}}+\gamma_{jb}+\epsilon_{ijb}$$
with ${\hat{y}}_{ijb}$ being the estimated abundancy for feature *j* and sample *i* in batch *b*. $x_{i}^{{\mathrm{LV}}_{k}}$ indicates the covariate (score) of the *k*th Latent Variable (LV) of sample *i*. $\gamma_{jb}$ is the (random) batch intercept for feature *j*. Note that, when the LV’s are sufficient in explaining $y_{ijb}$, the random intercept $\gamma_{jb}$ will not contribute much. Before fitting the model, we removed the outlier samples per batch *b* and feature *j* based on their within-batch Z-score (|Z|>2) determined from all samples in that batch. Note that these Z-scores are different from the Z-scores that are defined in other parts of this study.

The batch corrected abundancy were given by:(3)$$y_{ijb}^{\mathrm{batch}\;\mathrm{corrected}}\;=\;y_{ijb}-{\hat{y}}_{ijb}+\mathrm{Median}\left({\hat{y}}_{.jb}\right)$$

#### 4.4.3. Normalization by Best Correlated IS

The internal standard, *m*, which best correlates with a feature *j* is being used to normalize the abundancy of feature *j*. The correlation was measured within each batch while using the Spearman correlation between feature *j* and each internal standard individually across all samples and subsequently averaged across all nine batches. The internal standard that (positively) correlated the best was used for normalization according:(4)$${\hat{y}}_{ij}\;=\;\frac{y_{ij}}{y_{im}}\mathrm{Median}\left(y_{.m}\right)$$
with *m* being the best correlated internal standard.

#### 4.4.4. Normalization Methods from Literature

The following normalization methods were used in this study:
**Anchor** [6]: *Anchor* assumes a linear response between the features in the anchor samples and samples in the batch. An anchor sample is a fixed sample, which is analyzed in all nine batches, and it was included more than four times in each batch. Normalization was performed per batch by dividing each feature by the average of the anchor samples for that same feature per batch. In this study, we used our QC samples as the anchor samples.**CCMN** [15]: we used function normFit from the crmn R package with input argument ‘crmn’. As a design matrix, we chose QC samples versus human plasma’s.**EigenMS** [16]: QC samples and human plasma samples were treated as two different groups.**Fast Cyclic Loess** [17]: we used the normalize
CyclicLoess function from the limma R package while using the method ‘fast’ and iterations=100.**NOMIS** [18]: we used the function normFit from the crmn R package with input argument ‘nomis’.**PQN** [19]: *PQN* was implemented, as described by Filzmoser et al. The reference spectrum was given by the median of every feature j.**RUV** [20]: we used the function RUVRand from the MetNorm R package.**VSN** [21]: we used the vsn R package while using the vsn2 function.  

Some settings were optimized; the reader is referred to Section 4.4.6 for more details.

#### 4.4.5. Evaluation of Normalization Methods

Six metrics were used in order to evaluate the performance of normalization methods.

**WTR score**: the WTR score (Within variance Total variance Ratio) calculates the ratio between the ‘overall’ within-batch variance and the total variance from the QC samples:(5)$${\mathrm{WTR}}_{j}\;=\;\frac{\sigma_{j,\mathrm{within}}^{2}}{\sigma_{j,\mathrm{tot}}^{2}}\;=\;\frac{\sigma_{j,\mathrm{tot}}^{2}-\sigma_{j,\mathrm{between}}^{2}}{\sigma_{j,\mathrm{tot}}^{2}}$$
where $\sigma_{j,\mathrm{between}}$ is the variance of all nine batch averages for metabolite *j* in the QC samples, and $\sigma_{j,\mathrm{tot}}$ the ‘overall’ variance based on all QC samples. The WTR score is between 0 and 1. Because we would like batch averages to be similar for the QC samples (resulting in $\sigma_{j,\mathrm{between}}$ approaching zero), we are interested in WTR scores close to one. Note that the coefficient of variation (CV) was considered to be an inadequate metric, as a simple log-transformation of the data already results in a decreased CV. Because the WTR score considers a ratio between two standard deviations, this metric is less sensitive to such initial data transformations. **QC correlations**: for all QC samples, the Spearman correlations were calculated on the (normalized) abundancies. Normalization should increase the resemblance of the QC samples among each other, therefore increasing the Spearman correlations. It is expected that the Spearman correlations decreases when variations other then techical variation are removed.**Spearman score**: for the set of 15 quantitatively measured metabolites, we calculated the Spearman correlation between their quantitative measurements and the normalized abundancies. The overall normalization performance could be judged based on the median Spearman score of these 15 scores, having scores ∈[−1,1]. Higher values indicate better resemblance with the quantitative measurements.**$R^{2}$ score**: the $R^{2}$ between the quantitative measurements and the normalized abundancies of the 15 quantitatively measured metabolites. The overall performance could be judged from the median $R^{2}$ score, with scores of ∈[0,1]. Higher values indicate better (linear) fits with the quantitative measurements.**QC prediction score**: since the QC samples were different from the human plasma samples in terms of concentrations for several metabolites/features, we expect this difference to be observed in the first few principal components (PCs) of a Principal Component Analysis (PCA) analysis applied to all features (excl. standards). We fitted a logistic function while using the first four PCs as covariates and with class labels: ‘human plasma’ and ‘QC’. The fitted model returns per sample a probability of belonging either to the class ‘human plasma’ or ‘QC’. The probabilities for all samples are averaged into the QC prediction score. Increasing normalization performances should result in higher scores, as QC- and human plasma samples should be segregated. We used LogisticRegression from the Python package scikitlearn with parameters penalty=‘l1’, solver=‘saga’, multi_class=‘auto’, and max_iter=10,000 [22].**Batch prediction score**: increasing normalization performances should result in less batch clustering when examining the first few PCs of the PCA analysis (see QC prediction score). We fitted a logistic function for each batch versus all other eight batches while using the first four PCs as covariates and obtained the probability scores for all human plasma’s having the correct batch label. These scores were than averaged for all human plasma samples into a batch prediction scores ∈[0,1]. Scores that are closer to 1 indicate decreased normalization performances, since batch separation is (still) present.

#### 4.4.6. Settings for Normalization Methods from Literature

Based on the *Batch prediction score* and *QC prediction score*, we optimized the parameter settings for the following normalization methods: *CCMN*
ncomp = 8 for both ion modi, *EigenMS*
eigentrends = 6 for positive ion mode and eigentrends = 4 for negative ion mode, *RUVrand*
k = 5 for positive ion mode and k = 4 for negative ion mode and *Fast Cyclic Loess*
span = 0.6 for positive ion mode and span = 0.4 for negative ion mode. The reader is referred to Figure 5a–d for clarification of these choices.

### 4.5. Methods to Determine Metabolic Abberations

#### 4.5.1. Outlier Removal

In this study, we used reference samples (controls and patient) to calculated Z-scores. In order to prevent outlier samples (for a given feature/metabolite) to affect the accuracy of the Z-score, we used an iterative procedure to remove outliers before determining the set of samples used for calculating the Z-score. In this procedure, an ‘outlier Z-score’ was determined based on all of the samples (which samples were taken depends on the given Z-score method, see below), where samples having a |Z−score|>3 were removed. This was repeated five times and, from the non-outlier samples, a selection was made, depending on the selected Z-score method i.e., *15in*, *15out*, *All samples*, and *Regression* (see below).

#### 4.5.2. Z-Score Methods

Four different methods were used to determine Z-scores.

**15in**, best matching samples within batch: the Z-scores were calculated by selecting 15 samples originating from the same batch that were matched with the patient based on age and sex, as described in Bonte et al. [5].**15out**, best matching samples from other batches: the Z-scores were calculated similarly as in method *15in* while using 15 out-of-batch samples. Note that since there are more out-of-batch samples than within-batch samples the age and sex matching can be done more accurate for *15out* than for *15in*.**All samples**: this method used all available reference samples from all nine batches, including within-batch controls, for Z-score calculation, thereby ignoring age- and sex matching.**Regression**: we fitted a linear model on all available reference samples excluding outliers that were first removed based on a within-batch |Z−score|>3. This Z-score is different from other Z-scores mentioned in this study, and it is only used to remove outliers. The regression model is given by:(6)$$\begin{array}{cc} {\hat{y}}_{i} & ={\hat{\beta }}^{\mathrm{Intercept}}+{\hat{\beta }}^{\mathrm{Sex}}\;x_{i}^{\mathrm{Sex}}+{\hat{\beta }}^{\mathrm{Sex},\mathrm{Age}}\;x_{i}^{\mathrm{Sex}}x_{i}^{\mathrm{Age}}\\ \; & +\sum_{p=1}^{P}{\hat{\beta }}_{p}^{\mathrm{Age}}\;{\left(x_{i}^{\mathrm{Age}}\right)}^{p}+{\hat{\epsilon }}_{i}\\ {\hat{y}}_{i} & ={\vec{x}}_{i}^{T}\vec{\hat{\beta }}+{\hat{\epsilon }}_{i} \end{array}$$
where ${\hat{y}}_{i}$ is the predicted (normalized) abundancy of feature *j* for sample *i*, ${\hat{\beta }}^{\mathrm{Intercept}}$ is an intercept. ${\hat{\beta }}^{\mathrm{Sex}}$, ${\hat{\beta }}^{\mathrm{Sex},\mathrm{Age}}$ (interaction) and ${\hat{\beta }}_{p}^{\mathrm{Age}}$ indicate slopes. *P* is the degree of the polynomial used for regression on age and set to P=3 in this study. $x_{i}^{\mathrm{Sex}}$ is 1 for women and 0 for men. ${\hat{\epsilon }}_{i}$ is the estimated error. The latter expression is the model in vector notation with ${\vec{x}}_{i}^{T}\;=\;\left[1,x_{i}^{\mathrm{Sex}},\ldots ,{\left(x_{i}^{\mathrm{Age}}\right)}^{P}\right]$.

The coefficients were determined from the OLS estimator:(7)$$\vec{\hat{\beta }}\;=\;{\left(X^{T}X\right)}^{-1}X^{T}\vec{y}$$
where the rows of X are given by ${\vec{x}}_{i}^{T}$. The variance in ${\hat{y}}_{i}$ is determined by the variance in $\vec{\hat{\beta }}$ and the variance in ${\hat{\epsilon }}_{i}$:(8)$$\begin{array}{cc} \mathrm{Var}[{\hat{y}}_{i}] & =\mathrm{Var}\left[{\vec{x}}_{i}^{T}\vec{\hat{\beta }}\;\right]+\mathrm{Var}\left[{\hat{\epsilon }}_{i}\right]\\ \; & ={\vec{x}}_{i}^{T}\mathrm{Cov}\left[\vec{\hat{\beta }}\;\right]{\vec{x}}_{i}+{\hat{\sigma }}_{i}^{2} \end{array}$$

The covariance matrix of $\vec{\hat{\beta }}$ is given by:(9)$$\begin{array}{cc} \mathrm{Cov}[\vec{\hat{\beta }}] & =\mathrm{Cov}[\beta +{\left(X^{T}X\right)}^{-1}X^{T}\vec{\epsilon }\;]\\ \; & ={\left(X^{T}X\right)}^{-1}X^{T}\;E\left[\vec{\epsilon }\;{\vec{\epsilon }}^{T}\right]\;X{\left(X^{T}X\right)}^{-1} \end{array}$$

We estimated $E[\vec{\epsilon }\;{\vec{\epsilon }}^{T}]$ by:(10)$$\begin{array}{c} E\left[\vec{\epsilon }\;{\vec{\epsilon }}^{T}\right]\;=\;\left[\begin{array}{cccc} {\hat{\sigma }}_{1}^{2} & 0 & \cdots & 0\\ 0 & {\hat{\sigma }}_{2}^{2} & \cdots & 0\\ \vdots & \vdots & \ddots & \vdots \\ 0 & 0 & \cdots & {\hat{\sigma }}_{N}^{2} \end{array}\right] \end{array}$$

Because we expected $\sigma_{i}^{2}$ to be dependent on age (neglecting sex), we estimated ${\hat{\sigma }}_{i}^{2}$ from a weighted mean on the squared residuals:(11)$$\begin{array}{cc} {\hat{\sigma }}_{i}^{2} & =\sum_{k=1}^{N}\frac{w_{k}\left(x_{i}^{\mathrm{Age}}\right)}{\sum_{k^{\prime }=1}^{N}w_{k^{\prime }}\left(x_{i}^{\mathrm{Age}}\right)}{\left(y_{k}-{\hat{y}}_{k}\right)}^{2}\\ w_{k}\left(x_{i}^{\mathrm{Age}}\right) & \;=\;\exp\left(-\frac{|x_{i}^{\mathrm{Age}}-x_{k}^{\mathrm{Age}}|}{a+bx_{i}^{\mathrm{Age}}}\right) \end{array}$$
where *a* and *b* determine how the weights decay (*a*) or increase (*b*) over age (we set a,b=1 years). The Z-scores were obtained by subtracting the predicted average ${\hat{y}}_{i}$ and dividing by the variance $\mathrm{Var}[{\hat{y}}_{i}]$ (Equation (Equation 8)).

The significance of the regression coefficients (Equation (Equation 6)) was obtained by considering the statistic:(12)$$\frac{\left({\hat{\beta }}_{i}-\beta_{i}\right)}{\sqrt{\mathrm{Var}[{\hat{\beta }}_{i}]}}∼N\left(0,1\right)$$

The variances of the coefficients were found in the diagonal elements of $\mathrm{Cov}[\vec{\hat{\beta }}]$ (Equation (Equation 9)). We tested the hypotheses that $\beta_{i}=0$ with a two-tailed test. A robust *p*-value was obtained from a bootstrap procedure by taking the median *p*-value from a series of *p*-values that were obtained from 50 bootstraps on the above test statistics taking 90% of the data each bootstrap.

#### 4.5.3. Final Z-Scores

Because the patient samples were measured in triplicate, we determined the final Z-scores from the average of these three Z-scores [5]. These average Z-score were determined for all Z-score methods i.e., *15in*, *15out*, *All samples*, and *Regression*.

#### 4.5.4. *p*-Values from Welch’s *T*-Test

As an alternative to using the (average) Z-scores, we also considered the *p*-values that were obtained from the Welch’s *t*-test, as it indicates whether the mean of triplicates differs significantly from the population average. Note that the triplicate was expected to only have technical variance, whereas the reference population has variance that consists of technical- plus biological variance. For each Z-score method (*15in*, *15out*, *All samples*, and *Regression*), these *p*-values were obtained per feature (and patient).

When using the regression model, we used an adjusted Welch’s *t*-test assuming that the variance in the estimate of the average of the population (which is Z=0) was negligible:(13)$$t_{j}\;=\;\frac{\mathrm{Mean}(Z_{j.})}{\sqrt{\frac{s_{j}^{2}}{3}}}$$
where $s_{j}$ is the sample standard deviation of the triplicate Z-scores, $\mathrm{Mean}(Z_{j.})$ indicates the average of the triplicate for feature *j*.

### 4.6. Detection of the Expected IEM Biomarkers

In order to explore how normalization and the method of determining these Z-scores (*15in*, *15out*, *All samples*, and *Regression*) affected the detection of the expected biomarkers, we plotted the number of abberant biomarkers of the known IEM patients against the average number of abberant features (true plus false positives) per patients for various (final) Z-score and *p*-value cutoff levels, similar to a ROC curve. Improved biomarker detection was believed to increase the area under the curve (AUC).

Establishing this curve was done by assigning a status for every biomarker (if present and annotated in the MS-data). A database was established containing the expected biomarkers for each IEM, including the expected Z-score sign (up or down regulated), as can be found in Appendix D
Table A6. For every IEM patient, we assigned, for all expected biomarkers, the status ‘positive’ or ‘negative’. The status ‘positive’ was assigned when (1) $|Z-\mathrm{score}|>Z_{\mathrm{abnormal}}$ and (2) the sign of the Z-score corresponded with the expected sign for that biomarker in the IEM patient. When creating this curve while using the *p*-values, we also required that the sign of the Z-score corresponded with the expected sign for that biomarker, and similarly assigned the ‘positive’ status when $p-\mathrm{value}<p_{\mathrm{abnormal}}$. When a biomarker was found in both positive and negative ion mode, the Z-score(s) from the mode having the largest population average abundancy was taken. The average number of detected features (per patient) was obtained by considering the features from both ion modes.

Because some biomarkers are only found in a single IEM patient and not in reference samples (or other IEM patients), some of the expected IEM biomarkers were not matched across all nine batches and, therefore, were absent in the merged dataset and analysis in this study. In the merged dataset, we obtained 195 patient-biomarker combinations (one patient could have multiple biomarkers) that were associated with 49 patients.

## 5. Conclusions

In conclusion, out of all explored normalization methods, the removal of batch effects was best performed by *Log-Metchalizer*. Fitting our regression model on the corresponding normalized data showed that 6.5–37% (Table 1) of all considered features were dependent on age, underlining the need for using age corrected Z-scores. On average, biomarker detection performance using *Log-Metchalizer* using out-of-batch controls was at least similar to the best performing *Log-Raw* approach when using the 15 within-batch controls (*15in&Log-Raw*). We anticipate that the success of *Metchalizer* and age- and sex correcting strategies, such as our regression model, depend on three factors: (1) a feature of interest being measured in a number of other batches (not necessarily all), (2) batch effects containing (only) technical variations, and (3) abundancies being affected by age or other covariates and their effect size. In summary, our proposed approach using out-of-batch reference samples opens new opportunities for improving abnormality detection, especially for age-dependent features/biomarkers.

## Figures and Tables

**Figure 1 metabolites-11-00008-f001:**
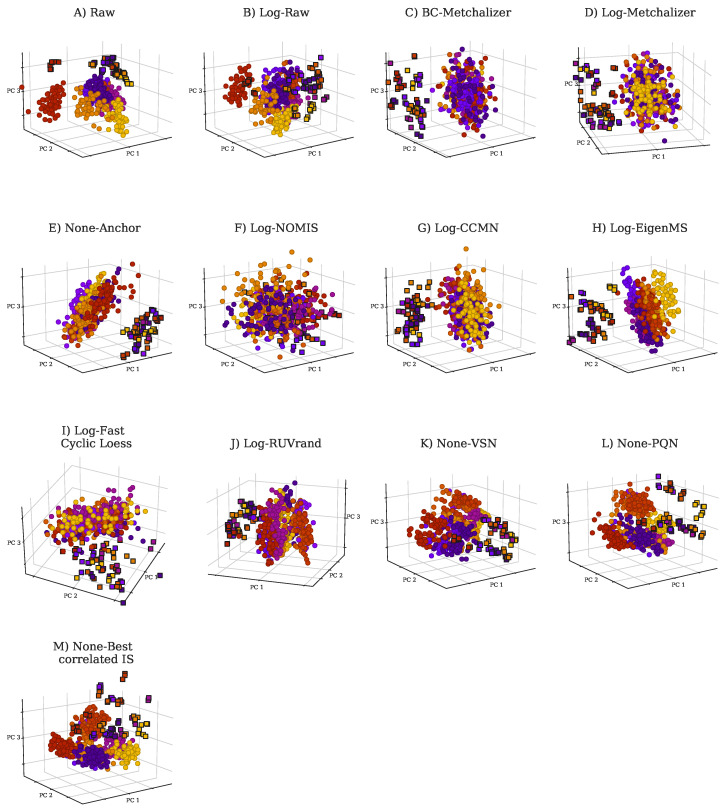
Principle Component Analysis (PCA) plots for raw data and normalized data as indicated by the title of each panel. Each batch is indicated with a unique color. PCA was performed on 431 features (excluding the internal and external standards) in positive ion mode. The squares indicate QC samples, whereas the circles indicate patient and control samples. (**A**) PCA plot for *Raw*, (**B**) *Log-Raw*, (**C**) *BC-Metchalizer*, (**D**) *Log-Metchalizer*, (**E**) *None-Anchor*, (**F**) *Log-NOMIS*, (**G**) *Log-CCMN*, (**H**) *Log-EigenMS*, (**I**) *Log-Fast Cyclic Loess*, (**J**) *Log-RUVrand*, (**K**) *None-VSN*, (**L**) *None-PQN*, (**M**) *None-Best correlated IS*.

**Figure 2 metabolites-11-00008-f002:**
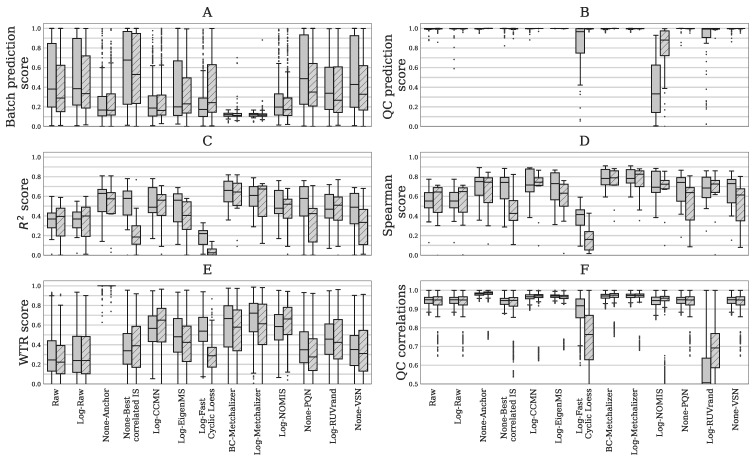
Six different performance metrics for batch effect removal (Section 4.4.5). Data from positive – or negative ion mode is indicated by plain and striped boxplots, respectively. (**A**) *Batch prediction score* measures the presence of batch effects in the first four principal components (PCs) from PCA analysis. (**B**) Quality Control (QC) prediction score measures how well QC samples are separated from human plasma sample in the first four PCs. (**C**) $R^{2}$ score between (normalized) abundancies and quantitative measurements. (**D**) *Spearman score* of (normalized) abundancies with quantitative measurements. (**E**) The *WTR score* measuring the overall within batch variation with respect to the total variance using the QC samples. (**F**) *QC correlations* measuring the resemblance of all QC samples among each other. Each data point represents a pair-wise Spearman correlation between two QC samples.

**Figure 3 metabolites-11-00008-f003:**
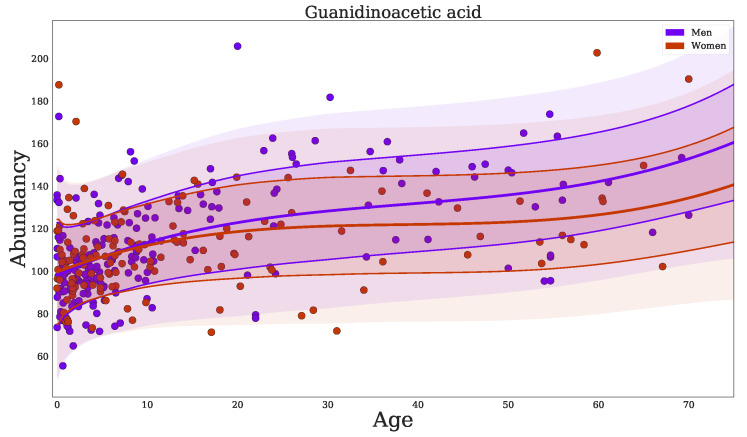
Regression of guanidinoacetic acid when using *BC-Metchalizer* normalized data. The different colors indicate the sex as shown in the legend. The thick red/blue line indicates the average obtained from the fit on all samples for a given sex. The first standard deviation is indicated by the thin(ner) line whereas the second standard deviation ends at the shaded region.

**Figure 4 metabolites-11-00008-f004:**
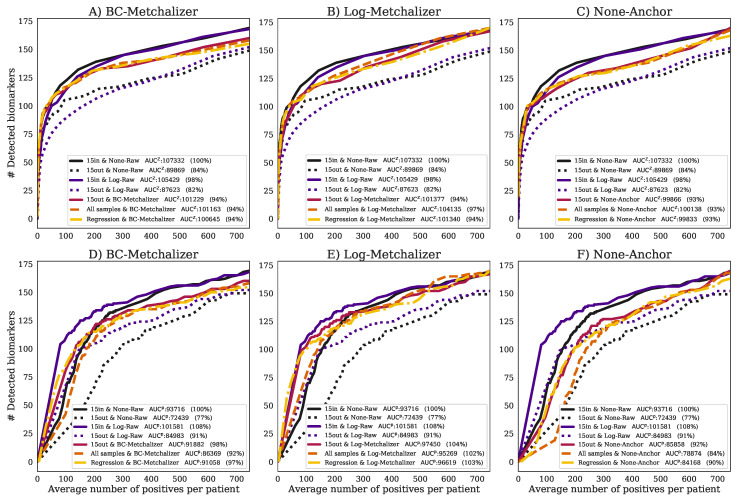
The number of detected expected biomarkers versus the average number of positives per patient. A curve in each (sub)figure was formed by increasing the Z-score or *p*-value threshold ($Z_{\mathrm{abnormal}}$, Methods). The legend indicates (per curve) the methods used to determine Z-scores and how data was normalized, the AUC and AUC expressed as percentage of the ${\mathrm{AUC}}_{15\mathrm{in}\&\mathrm{None-Raw}}^{x}$. Performances using (**A**) *BC-Metchalizer* and Z-scores, (**B**) *Log-Metchalizer* and Z-scores, (**C**) *None-Anchor* and Z-scores, (**D**) *BC-Metchalizer* and *p*-values, (**E**) *Log-Metchalizer* and *p*-values, and (**F**) *None-Anchor* and *p*-values.

**Figure 5 metabolites-11-00008-f005:**
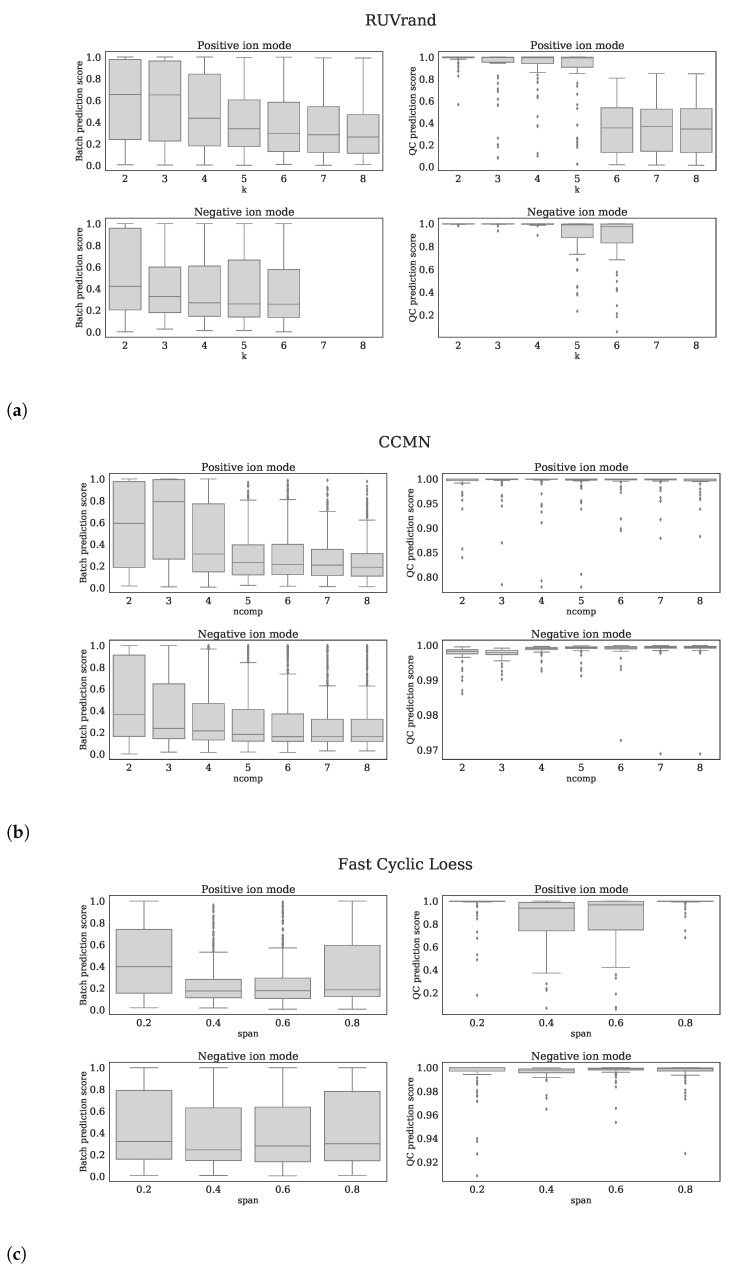
Each subfigure shows 4 panels which belong to the normalization method stated in the title. The upper, lower panels are the *batch prediction scores* and *QC prediction scores* (vertical axes) for various choices of the parameter (horizontal axis) for positive, negative ion mode, respectively. (**a**) *RUVrand*, (**b**) *CCMN*, (**c**) *Fast Cyclic Loess*, (**d**) *EigenMS*.

**Table 1 metabolites-11-00008-t001:** The percentage of significant coefficients in the regression model for a given ion mode and normalization method.

Coefficient	Ion Mode	None-Anchor	BC-Metchalizer	Log-Metchalizer
${\hat{\beta }}^{\;\mathrm{Intercept}}$	−	98.4	100.0	100.0
${\hat{\beta }}^{\;\mathrm{Intercept}}$	+	100.0	100.0	100.0
${\hat{\beta }}^{\;{\mathrm{Age}}_{1}}$	−	22.9	36.9	31.8
${\hat{\beta }}^{\;{\mathrm{Age}}_{1}}$	+	6.5	12.5	13.2
${\hat{\beta }}^{\;{\mathrm{Age}}_{2}}$	−	4.5	17.8	16.9
${\hat{\beta }}^{\;{\mathrm{Age}}_{2}}$	+	0.5	4.4	5.1
${\hat{\beta }}^{\;{\mathrm{Age}}_{3}}$	−	1.6	8.3	8.3
${\hat{\beta }}^{\;{\mathrm{Age}}_{3}}$	+	0.2	0.2	0.5
${\hat{\beta }}^{\;\mathrm{Sex}}$	−	0.0	0.0	0.0
${\hat{\beta }}^{\;\mathrm{Sex}}$	+	0.0	0.0	0.0
${\hat{\beta }}^{\;\mathrm{Sex},\mathrm{Age}}$	−	0.0	0.0	0.0
${\hat{\beta }}^{\;\mathrm{Sex},\mathrm{Age}}$	+	0.5	0.7	0.0

## Data Availability

The data presented in this study are available in this article.

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
