# Peer review of "Using Out-of-Batch Reference Populations to Improve Untargeted Metabolomics for Screening Inborn Errors of Metabolism"

_metabolites, 2020, doi:10.3390/metabo11010008_

Round 1

Reviewer 1 Report

Bongaerts et al investigated methods to merge and accurately normalize data from multiple batches to apply metabolomics in IEM diagnostics. The authors state that this is urgently needed, which I completely support. The developed script "Metchalizer", applies normalization and takes the effects of age (and gender) into account, which seems to allow to accurate diagnose disease without using multiple controls per run.

Major points:

  1. From the manuscript, it is not clear whether Methalizer will be open source software after publication. Is this the intention? I recommended the editor to ask a bioinformatician to thoroughly check the formulas and test Mechalizer before publication.
  2. Pg 3, line 78: The authors have included 9 untargeted metabolomics runs/batches to study the effect of different normalization procedures. Are specific sources of variation included to study the effect of normalization on between-batch variation? The main question behind it is whether Metchalizer will correct for large batch effects in intensity (machine dependent?), retention time shifts by changing columns, eluentia etc. If so, which sources?
  3. Pg 3, line 82: The authors state that they “only included features which were merged across all nine batches to ensure consistency among the findings. This resulted in the loss of some IEM biomarkers”. How many were lost? And did you lose important biomarkers leading to low diagnostic accuracy on a specific disease, and if so, which disease?

Other points

  1. Pg 2, line 35: “When using untargeted metabolomics the establishment of reference values is complicated due to the semi-quantitative nature of the data owing to several sources of variation like injection volume, retention time, temperature, or ionization efficiency …. Please consider adding that internal standards for every component cannot be added to control for these sources of variation.
  2. Pg 3, line 100: QC samples: the atuhors have applied a single QC commercial sample. The samples are very different from the patient samples, as shown in fig 1. Please comment on why only 1 sample and a commercial (less presentative) sample was chosen? Could the choice of QC affect the outcome of Metchalizer?
  3. Pg 3, line 101: is separate a correct term?
  4. Pg 4, fig 1. The y-axes of fig 1-I and 1-J are merged.
  5. Pg 6, line 169: two typos: positive => positive and interection => interaction.
  6. Pg 11, line 327: 18-40 control samples were analyzed within each batch; were similar samples analysed during each run, or did they differ between runs? Could this affect the performance of Metchalizer?
  7. Pg 12, line 357: Not sure what is meant here: “As we expect matching features to have similar within-batch median abundancies (despite of batch effects), we calculated the differences between these medians in percentages, which had to be < 300%.” Do you mean that feature abundancies (normalized?) BETWEEN-RUN cannot exceed >300%? Does this not depend on the samples included within the batches? Please clarify.
  8. Pg 12, line 364: Not sure what is meant here: Features matching multiple other features in the reference batch were discarded (and vice versa). As you intend to include features that are matched across all batches, this does not seem logical. Or did you mean features that were matched only in the reference batch and visa versa?
  9. Pg 17, line 479: Criteria 1 (p-value instead of Z-score) and 2 were also used for the curve created by the p-values. What is criterion 2? It does not seem to refer to the text at bullet 2 in the previous sentence.
  10. Pg 19, Table A1: It would be interesting to show the post-normalization CV as well, using Methalizer.
  11. Pg 21, Figure A2: please explain how the information in this Figure adds to the manuscript. It is not clear from the legend and the main text.

Reviewer 2 Report

Outlier or out-of-batch samples are important for research use, but due to lack of suitable handling tool it is often removed from the analyzed dataset. Here, Bongaerts et al developed Metchalizer a normalization method enables the use of out-of-batch reference samples to establish clinically relevant reference values for metabolite concentrations. Moreover, this study focused Untargeted metabolomics as a promising alternative that may enable the determination of many metabolites in one analysis and overcome the limitation of targeted metabolomics.

Major concern: 

1. Author mostly choose technical replicates for making their batches and looks like justifying Metchalizer as the best approach for out-of-batch samples normalization. Author must use biological replicates based batches as this is the standard practice for global metabolomics analysis. 

2. How author claim that their untargeted metabolomics approach overcome the limitations of the targeted metabolomics is obscure. for example, how relevant of qualitative metabolomics to targeted (quantitative) metabolomics? 

3. Title also ambitious. 

Reviewer 3 Report

The manuscript describes a study on out-of-batch reference samples for metabolomics in a clinical setting. They are attempting to tackle an important and highly relevant problem that many laboratories are facing.

I have a few remarks to improve and/or clarify the manuscript.

  • If samples of the same controls/patient are run in multiple runs, can the authors show that patients cluster closer to their own data, rather than patients cluster with patients and not QCs? You want to maintain relevant differences between patients
  • Similarly, since authors note that the constitution of the QC is quite different from other samples, why optimize based on separation between QC and samples rather than separation of clinically/biochemically relevant groups?
  • Why are patients analyzed with technical triplicates and other controls not?
  • Why are there 5-9 technical triplicates of QC samples?
  • Based on the plot of fig 2F, there seem to be quite a few outliers. Do the authors have an explanation? And it is not yet clear to me what an individual data point is here, expanding the legend may help.
  • Line 324 Commercial QC monster ; which firm, which components are present?
  • Line 333 what are concentrations of the internal standards?

Line 26 GS-MS -> should it read GC-MS?

Line 78 define “merged”

Line 79 patients -> should read patient samples

Line 81 merged should be present?

Fig 1 left Y-axis: C should be PC

Spelling mistake in correspondence email address

Round 2

Reviewer 2 Report

None